# Considerations for Augmenting Aripiprazole Long-Acting Injectables with Other Antipsychotics: A Mini-Review

**DOI:** 10.3390/diseases13080274

**Published:** 2025-08-21

**Authors:** Jonathan Shaw, Ethan Kim, Emily Ton, Charles Lai, Peter Bota, Tina Allee

**Affiliations:** 1School of Medicine, California University of Science and Medicine, Colton, CA 92324, USA; 2The Internal Medicine Service, Loma Linda University Health, Loma Linda, CA 92354, USA; 3Inpatient Psychiatric Facility Service, College Medical Center, Long Beach, CA 90806, USA

**Keywords:** aripiprazole, long acting injectable, schizophrenia, psychopharmacology, antipsychotics

## Abstract

Aripiprazole is a third-generation antipsychotic, approved in 2002, notable for its partial agonism of the Dopamine D2 receptor and lower risk of metabolic and extrapyramidal adverse effects. It is available in a long-acting injectable formulation, which is very useful for maintaining medication compliance, which is crucial for preventing recurrent psychotic episodes in patients. Additionally, the aripiprazole long-acting injectable is frequently combined with other antipsychotic medications in acute settings to manage refractory symptoms. However, there is limited literature regarding the psychopharmacology, efficacy, and adverse effect profiles of augmenting aripiprazole long-acting injectable with other antipsychotic medications. This narrative review intends to synthesize the existing literature on aripiprazole, its comparative affinity to the dopamine D2 receptor versus other antipsychotics, and the efficacy and side effect profiles of combining aripiprazole with other antipsychotics in the context of acute inpatient treatment for psychosis. Current literature on K_i_ values indicates that fluphenazine, pimozide, thiothixene, trifluoperazine, and perphenazine bind more strongly to dopamine D2 receptors than aripiprazole. However, there is a knowledge gap regarding antipsychotic polypharmacy with aripiprazole and these first generation antipsychotics, limiting the discussion of these drug combinations to theory. Additionally, the muscarinic effects of aripiprazole suggest the possibility of augmentation with clozapine or xanomeline-trospium, albeit the peer-reviewed literature on this was also limited. Overall, it is difficult to draw conclusions regarding best clinical practices for these scenarios, as the existing literature is contradictory. Nonetheless, the application of the dopamine and muscarinic pathway theories for schizophrenia opens venues for future research and consideration.

## 1. Introduction

First approved for schizophrenia by the United States Food and Drug Administration (US FDA) in 2002, aripiprazole entered the market as the first of what would become known as the third-generation antipsychotics [1]. Aripiprazole is a partial dopamine D2 subtype (dopamine D2) agonist, which differs from first-generation antipsychotics (FGA) that mainly antagonize the dopamine D2 autoreceptor, and from second-generation antipsychotics (SGA) that interact with serotonergic receptors in addition to their dopamine D2 antagonism [1,2]. Aripiprazole has subsequently been approved as an orally disintegrating tablet, oral solution, and long-acting intramuscular injection [1]. Aripiprazole has been approved for schizophrenia in adults and adolescents (13–17 years old), bipolar 1 disorder (manic and mixed episodes) in adults and pediatric patients (10–17 years old), as an adjunctive therapy for major depressive disorder in adults, for irritability associated with autism spectrum disorder in pediatric patients (6–17 years old) and vocal/motor tics associated with Tourette’s syndrome in pediatric patients (6–18 years old) [1]. The intramuscular formulation has also been approved for agitation associated with schizophrenia or bipolar mania in adults [1].

Recent neuroimaging research has deepened our understanding of aripiprazole’s receptor binding profile and its clinical implications. A triple-tracer positron emission tomography study revealed that aripiprazole achieves high striatal dopamine D2 receptor occupancy, averaging 87–93%, without a proportional increase in extrapyramidal symptoms (EPS), which are typically seen with other high-occupancy antipsychotics like haloperidol [3]. This paradox is likely due to its unique partial agonist activity, which stabilizes dopaminergic tone rather than fully blocking receptor activity [2,3,4]. Furthermore, its moderate affinity for serotonin 5-hydroxytryptamine subtype-2A (5-HT2A) receptors and low affinity for 5-hydroxytryptamine subtype-1A (5-HT1A) receptors contribute to a differentiated clinical profile compared to other SGAs [3]. This pharmacological profile supports aripiprazole’s utility in patients who require sustained D2 occupancy but are at risk for side effects such as EPS or prolactin elevation.

The introduction of aripiprazole has also contributed to the rethinking of antipsychotic efficacy in terms of receptor binding kinetics and functional selectivity. Compared to typical antipsychotics that exhibit tightly bound D2 antagonism, aripiprazole’s rapid dissociation from the D2 receptor may allow for endogenous dopamine to compete more effectively, possibly resulting in a lower side effect burden [2,5]. Given the heterogeneity of schizophrenia and the growing interest in individualized, mechanism-based interventions, aripiprazole’s distinct properties make it a compelling candidate for investigation in complex treatment scenarios, such as hospitalized patients already receiving long-acting injectable (LAI) formulations who require rapid symptom stabilization. Due to the increased medication compliance and improved quality of life associated with LAI formulations compared to oral antipsychotics, the authors believe that augmenting LAI treatments is a worthwhile consideration when tailoring patient medications for their individualized presentation [6,7].

This narrative review intends to serve as an overview of the pharmacology of aripiprazole with focus on its effects on muscarinic pathways and the Dopamine D2 Receptor (DRD2), the implications of that pharmacology for antipsychotic polypharmacy in the treatment of inpatient high-acuity and treatment-resistant psychosis, and its clinical utility particularly as an LAI when used in conjunction with other antipsychotics.

## 2. Methods

To conduct this narrative review, the authors used various databases including PubMed, Scopus, Google Scholar, and Web of Sciences to find relevant literature. No specific set of search terms was used, but the authors generally searched for peer-reviewed sources that addressed the mechanisms of action of antipsychotics, their side effect profiles, and any relevant clinical trials associated with these medications. As this is a narrative review, no research protocol was registered.

All K_i_ values were sourced from the PDSP K_i_ database [8]. The following search terms were used in the database: receptor—dopamine D2; hot ligand—3H-NMSP; species—human; source—cloned. If multiple K_i_ values were found, the K_i_ values were provided as a minimum-maximum range and as an average. K_i_ values of <30 nM are high affinity, K_i_ values between 30–300 nM are moderate affinity, and K_i_ values >300 nM are low affinity.

## 3. Results

### 3.1. Mechanism Specifics

Aripiprazole is primarily known as a partial agonist of dopamine D2 and D3, along with partial agonism of serotonin 5-HT1A and an antagonism of serotonin 5-HT2 receptors [1,3]. Aripiprazole has a very high occupancy of striatal D2 receptors (average occupancy of 87% in the putamen, caudate 93%, and ventral striatum 91%), lower occupancy of serotonin 5-HT2 receptors (54–60%), and even lower occupancy of serotonin 5-HT1A receptors (16%) [3]. Studies have suggested that the occupancy threshold for clinical response for the use of aripiprazole in schizophrenia is above 80% for DRD2s, with only 2 out of 4 participants with occupancies exceeding 90% presenting with EPS as seen in Figure 1 [3,4]. Physiologically based pharmacokinetic models that correlate plasma concentration and simulated DRD2 occupancy with predicted/observed Richmond Agitation-Sedation Scale (RASS) scores have been developed for other antipsychotics, such as haloperidol [4]. The percentage of observed decrease in RASS scores strongly correlated with the predicted RASS decrease according to this model (Slope 0.997 ± 0.015; Pearson’s r = 0.999), demonstrating a strong association between dopamine receptor occupancy and the clinical effectiveness of antipsychotics [4,5]. A linear correlation between the odds ratio of EPS and peak D2LR antagonism as functions of dopamine receptor occupancy (Slope 0.144 ± 0.003; Pearson’s r = 0.959) for haloperidol was also noted, indicating that DRD2 occupancy must be considered when balancing clinical effectiveness with the risk of EPS and other side effects [2,4]. This is particularly important due to the relatively long half-life of aripiprazole at around 60–70 h, with some studies indicating that it can saturate dopamine D2/D3 receptors for up to a week [5].

Aripiprazole also has a high affinity to recombinant human serotonin 5-HT1A receptors (K_i_ = 1.65 nmol) and displays a similar level of potency and intrinsic activity to ziprasidone (Emax of 68% for aripiprazole, Emax of 79% for ziprasidone of 5-HT) [3]. This is important because partial agonism at the 5-HT1A receptor is one of the proposed ways in which aripiprazole alleviates depression, anxiety, negative symptoms, and EPS [3]. Aripiprazole is predominantly metabolized via phase I mechanisms equally by both Cytochrome P450 2D6 (CYP2D6) and Cytochrome P450 3A4 (CYP3A4), along with limited phase II metabolism [1,9]. The major circulating metabolite depends on the frequency of administration, as phase II product Bristol-Myers Squibb 337041st compound (BMS-337041), a metabolite of aripiprazole, is most abundant during acute dosing, while phase I product dehydroaripiprazole is more prevalent in chronic dosing [1]. Of note, if a patient with schizophrenia receives adjunctive antidepressants that are also metabolized by CYP2D6 and CYP3A4 (e.g., Fluoxetine, Paroxetine, etc.), increases in the serum concentrations of both the antidepressant and aripiprazole may occur [1,9].

The affinity constant values (K_i_) of each ligand are calculated from the Cheng-Prusoff equation (K_i_ = IC50/(1 + C/K_d_) where K_d_ is the dissociation constant, IC50 is the half-maximal inhibitory concentration, and C is the concentration of the ligand [10]. A lower K_i_ value indicates a greater ability to bind to receptors and is associated with a stronger pharmacological action [10]. This differs from the inhibition constant, indicated as pK_i_ (−log K_i_), which can be erroneously reported as K_i_ values when examining the literature [10]. This review will only use the K_i_ value for clarity, with K_i_ values of <30 nM considered high affinity, K_i_ values between 30–300 nM moderate affinity, and K_i_ values >300 nM low affinity [1]. The average DRD2 K_i_ values of various antipsychotics can be seen in Table 1. At the time of writing this review, the authors were unable to find any peer-reviewed DRD2 K_i_ values for dehydroaripiprazole and BMS-337041.

### 3.2. Side Effects Profiles of the Various Generations of Antipsychotics

FGAs are associated with EPS, such as movement disorders (e.g., dystonia, akathisia, parkinsonism, tardive dyskinesia) and other disorders like hyperprolactinemia [1,2,13]. EPS are caused by blockade of DRD2 in the nigrostriatal pathway, while hyperprolactinemia is caused by blockage of DRD2 in the tuberoinfundibular pathway [1,13]. SGAs are as clinically effective as FGAs for medical management of psychosis with lower incidences of EPS and hyperprolactinemia, likely because SGAs occupy less than 60% of striatal DRD2 (as opposed to FGAs, which can occupy 65%) and/or the faster dissociation from DRD2 binding sites [1]. The antagonism of serotonin 5-HT2A receptors by SGAs has also been proposed to reduce the need for more extensive DRD2 occupancy, leading to reduced side effects [1]. However, this serotonin antagonism hypothesis is challenged by the fact that haloperidol and other FGAs have comparable antagonistic properties for the 5-HT2A receptors [1].

Aripiprazole’s unique receptor binding profile contributes to a side effect profile that differs significantly from both FGAs and many SGAs. As a partial agonist at the dopamine D2 receptor, aripiprazole can act as either an antagonist or agonist depending on the endogenous dopamine tone, which may explain its lower propensity to cause EPS at therapeutic doses despite high DRD2 occupancy [2,3]. The most commonly reported side effects of aripiprazole are akathisia in adults and tremors in adolescents with schizophrenia [1]. Of note, there is a black box warning for adjunctive treatment with antidepressants in patients 24 years and younger for an increase in suicidal thoughts and behavior [1]. Other undesired side effects include dizziness, drowsiness, sedation, insomnia, somnolence, weight gain, drooling, restlessness, anxiety, and headaches [1,9]. Furthermore, aripiprazole exhibits low affinity for histamine subtype H1 (H1) and muscarinic receptors, contributing to a reduced risk of sedation, weight gain, and anticholinergic effects compared to other antipsychotics like olanzapine or clozapine [1]. Nonetheless, individual susceptibility to side effects remains variable, and clinicians should monitor for behavioral activation, sleep disturbances, and metabolic parameters, especially in vulnerable populations such as adolescents or patients with affective symptoms.

### 3.3. Use with Other Antipsychotics and Psychotropic Medications

Aripiprazole is primarily metabolized by Cytochromes P450 (CYP450) enzymes such as CYP2D6 and CYP3A4 [1,9]. Drug–drug interactions with applicable Cytochromes (CYP) inducers such as carbamazepine and inhibitors such as fluoxetine, quinidine, or ketoconazole should be considered when determining dosings for aripiprazole in patients with psychiatric comorbidities [9]. More information regarding specific dosage adjustments is presented in Table 1. At the time of writing this narrative review, the authors were unable to find literature definitively discussing the augmentation of aripiprazole LAI with an additional antipsychotic for acute psychosis, paranoia, and suicidality in an inpatient setting. Theoretically, based on the K_i_ values for the DRD2, fluphenazine, pimozide, thiothixene, trifluoperazine, and Perphenazine are antipsychotics that may bind more strongly than aripiprazole [8,11,12]. In terms of clinical application, the use of dual LAI in treatment-resistant schizophrenia for patients who refused or were otherwise unable to tolerate clozapine has been explored in the literature, albeit published data on this topic remains limited, and antipsychotic polypharmacy (APP) is a contentious topic in the literature [13,14,15]. For context, a 2024 systematic review included 15 relevant studies comprising 9 case reports, 4 case series, and 2 observational retrospective studies for a total sample size of 123 patients, all of whom were diagnosed with schizophrenia or schizoaffective disorder, except for 12 patients who were primarily diagnosed with bipolar disorder, organic diseases, and intellectual disability [14]. This relatively small sample size warrants caution when interpreting the results of this literature review for implementation into clinical practice. However, their findings still provide valuable insights when considering medication management in such scenarios.

Current literature presents a wide variety of dual LAI antipsychotic combinations in clinical use, with 12 different combinations of FGA and SGA LAI, 5 combinations of two different SGA LAI, and 4 combinations of two FGA LAI [14]. The most commonly used FGAs for dual FGA LAI were haloperidol decanoate, zuclopenthixol decanoate, and flupenthixoldecanoate [14]. The most commonly used SGAs for dual SGA LAI were aripiprazole monohydrate and paliperidone palmitate [14]. Adverse events after dual LAI administration were found in four patients (3.25%), including unspecified EPS, mild bradykinesia, polydipsia, and polyuria [14]. Additionally, one of the observational retrospective studies measured lab values and found no significant changes pre- and post-treatment, though the current literature disputes this finding, as some studies have found antipsychotic polypharmacy was more likely to elevate prolactin than antipsychotic monotherapy (APM) [14,15]. Despite these promising findings, the small sample size of 123 patients and relatively short duration of follow-up (ranging from 2 to 18 months) limit the ability to make clinical recommendations based on the existing literature [14]. Of note, there were no reported significant adverse events (SAE) from dual LAI in the reviewed literature [14]. SAE is defined as any untoward medical occurrence at any dose that results in death, is life-threatening, requires inpatient hospitalization or prolongation of existing hospitalization, results in persistent or significant disability, or causes a congenital anomaly or birth defect [16]. However, as there was no separate analysis for the dual LAI treatments involving aripiprazole, these findings cannot be generalized to aripiprazole specifically.

A 2019 meta-analysis of 42,600 patients on antipsychotics determined that patients on antipsychotics are at risk of at least one somatic SAE occurring to them compared to patients on placebo, with an odds ratio of 1.24 (95% CI 1.10, 1.41) [16]. The majority of the studies (88%) in this meta-analysis were considered short-term studies (duration of <13 weeks), and the authors were unable to analyze the remaining data to sufficiently identify the risk of somatic SAE in patients on long-term antipsychotics [16]. The lack of SAE found in the 2024 systematic review may be due to the small sample size of patients and literature, but it could also be due to differing definitions for adverse events [14]. Despite the lack of SAE, 3.25% (*n* = 4) of patients on dual LAI reportedly had adverse events, such as unspecified EPS, mild bradykinesia, polydipsia, and polyuria, which should be taken into consideration for APP [14].

Although it has been noted in literature that APP can produce better results than other treatment augmentations, such as benzodiazepines, existing meta-analyses have yielded mixed results, likely due to the small sample sizes of the current literature and the lack of separation of high and low quality studies in analyses [15]. Older reviews support APM, while newer reviews support APP [15]. Additionally, organizational guidelines for APP vary from recommending only monotherapy (American Psychiatric Association) to allowing additional antipsychotics to augment clozapine in cases where clozapine monotherapy is not effective (National Institute for Health and Care Excellence) [15]. The topic of APP is contentious in the literature, with some meta-analyses supporting its use with findings that APP has superior symptom reduction compared to APM and that transitioning from APP to APM is associated with an increased risk of medication nonadherence (RR 2.28, 95% CI 1.50–3.46) [15]. Meanwhile, other studies have found that patients recently discharged from hospital with APP have a significantly increased risk of readmission (HR 1.4, 95% CI 1.2–1.7) compared to APM patients, though it was noted that patients in the APP group for the cited study may have had more severe clinical manifestations which led to these findings [15]. Although it was previously thought that APP is more likely to elevate prolactin than APM, this notion has also been disputed in recent literature [14,15]. Therefore, it is difficult to draw definitive conclusions on best practices for APP and dual LAI use for treatment-resistant schizophrenia, but APM with either clozapine or non-clozapine antipsychotics should first be pursued due to the generally lower health service costs and risks of adverse effects [14,15]. Patients on APP should be carefully considered for individualized cases with close follow-up and recording of symptoms, since the literature indicates that many APP patients can safely transition to APM [15].

## 4. Discussion

### 4.1. Mechanistic Considerations

With the limited sample sizes and disputed findings in the existing literature, it is difficult to find conclusive recommendations for antipsychotic augmentation of aripiprazole LAI in the setting of acute psychosis. Many antipsychotics do not have peer-reviewed DRD2 K_i_ values or may differ substantially, since researchers use different hot ligands and species in their experiments. In this narrative review, the hot ligand of 3H-N-methylspiperone (3H-NMSP) and species of human-cloned was used as it provided the most comparable K_i_ values of various antipsychotics in the University of North Carolina at Chapel Hill K_i_ database. DRD2 K_i_ values for a given antipsychotic ranged significantly, so the range and average K_i_ value were taken into consideration with fluphenazine, pimozide, thiothixene, trifluoperazine, and perphenazine appearing to bind more strongly than aripiprazole to the DRD2 [8,11,12]. Since these are all FGA, their use may be limited due to their greater potential of inducing EPS and other associated side effects. Additionally, the authors were unable to find any K_i_ values for other TGA, such as brexpiprazole and cariprazine, in the UNC K_i_ database which limits their inclusion for this mechanistic discussion. However, clinical trials and systematic reviews involving TGAs like brexpiprazole have found that they have comparable efficacies in managing psychiatric symptoms to aripiprazole, though there is a slightly increased risk of weight gain and akathisia compared to aripiprazole, but this can be mitigated by initiating brexpiprazole at 0.5 mg/day before titrating to therapeutic levels [17,18]. Theoretically, switching from aripiprazole LAI to another TGA is viable for managing schizophrenia, but it should be noted that aripiprazole is the only TGA with an LAI option currently approved for the market [18]. This potentially limits the practical viability of APP using non-arpipirazole TGAs as combining oral antipsychotics which may have differing daily dosing schedules or requirements, e.g., the caloric requirement of lurasidone, can be difficult to track and frustrating for patients, inhibiting medication compliance [7].

Therefore, instead of solely considering DRD2 antagonism, other receptors should be considered when augmenting aripiprazole LAI use. For example, the combination of oral clozapine and aripiprazole has been brought up in the literature due to its potential to have limited EPS side effects, as well as one of its proposed mechanisms of action for improving cognition in patients with schizophrenia as a potent partial muscarinic acetylcholine receptor M1 (M1) receptor agonist [7,14,15,19,20,21,22]. Although the dopamine hypothesis is the primary theory through which we treat schizophrenia, other proposed neurotransmitter theories address the manifestations of various schizophrenia symptoms [19,22,23]. One major theory involves muscarinic acetylcholine receptors (mAChR) and can be visualized with Table 2, with particular emphasis on the M1 receptor [19,22,23,24]. M1 receptor agonists have been shown to enhance synaptic plasticity, increase neuronal excitability, and facilitate learning/improve cognition, especially when acting on the prefrontal cortex and hippocampus [19,22,24,25]. Prior clinical research into mAChR agonists, like arecoline, demonstrated brief, lucid intervals of improvement in psychotics in patients with schizophrenia [19,26]. M1 agonism facilitates excitatory postsynaptic currents, potentiating N-methyl-D-aspartate (NMDA) receptors that regulate inhibitory type A gamma-aminobutyric acid (GABA-A) receptors in dopaminergic neuron-regulated brain regions, thus treating schizophrenia through a more indirect pathway [19,23,26]. The major metabolite of clozapine, N-desmethylclozapine (NDMC), is a potential partial M1 receptor agonist and is thought to be a major contributor to why clozapine was able to improve cognition and working memory in patients with schizophrenia [19,22]. However, NDMC has no dopaminergic activity and is unable to effectively treat the positive symptoms of schizophrenia by itself; clozapine, with its DRD2 antagonism, also plays an active role in treating schizophrenia [19]. Instead, based on theoretical receptor-based rationale, NDMC is thought to primarily affect the cognitive symptoms of schizophrenia through its muscarinic activity as seen in preclinical models and Phase II clinical trials which report that lower clozapine to NDMC ratios are associated with improvements in working memory and executive function, whereas higher clozapine to NDMC ratios are associated with cognitive deficits [19,22].

Similarly, xanomeline-trospium (XT) treats schizophrenia through the muscarinic pathway [19,22,24,25,27,28]. Xanomeline, the main psychoactive component, agonizes M1 and muscarinic acetylcholine M4 (M4) muscarinic receptors, while trospium serves as a peripheral cholinergic antagonist [22,27,28,29]. M4 receptors are highly expressed in the dorsal striatum and nucleus accumbens [Table 2], within the nigrostriatal and mesolimbic dopaminergic pathways, respectively [25,29]. Agonism of M4 receptors is hypothesized to reduce hyperactivity in these dopaminergic pathways without directly antagonizing DRD2, thus producing antipsychotic effects while limiting EPS [25,28,29]. The Phase III Evaluating the Muscarinic Effects on Recognition and Global Enhancement in Negative Traits (EMERGENT) randomized clinical trials demonstrated that XT monotherapy, as compared to placebo, produced statistically significant reductions of Positive and Negative Syndrome Scale (PANSS) scores in patients with schizophrenia [28,30]. However, the phase III “A Study to Assess Efficacy and Safety of Adjunctive KarXT in Subjects With Inadequately Controlled Symptoms of Schizophrenia” (ARISE) trials evaluated XT as adjunctive therapy for patients on stable doses of risperidone, paliperidone, aripiprazole, ziprasidone, lurasidone, or cariprazine with inadequately controlled symptoms of schizophrenia and failed to show statistically significant improvement in PANSS scores (*p* = 0.11) [31,32]. Post hoc subgroup analysis of non-risperidone subgroups (including aripiprazole) demonstrated a statistically significant difference in PANSS scores (*p* = 0.03) [31], suggesting that augmenting aripiprazole LAI with XT may produce promising results [22]. Although data is still limited, the alternative pathways and mechanisms of aripiprazole and XT may demonstrate more tolerable side effect profiles (e.g., diarrhea, constipation, nausea, vomiting), with a lower risk of EPS [24,28,33].

It should be noted that, although XT is currently approved as a stand-alone therapy for the treatment of schizophrenia, augmentation of existing antipsychotic therapy with XT is not currently FDA approved. Likewise, clozapine is an FDA approved medication for treating schizophrenia while NDMC, a metabolite of clozapine, is not FDA approved as a stand-alone treatment for schizophrenia. This mechanistic discussion of how XT and NDMC could augment aripiprazole LAI through the muscarinic pathway is theoretical and not a clinical recommendation. NDMC does not appear to be under investigation for augmenting existing dopaminergic antipsychotics, but XT is currently undergoing clinical trials as an adjunctive treatment to SGA for adult patients with schizophrenia through the ARISE trial [31].

### 4.2. Patient Quality of Life and Logistical Considerations

Various factors, such as antipsychotic side effect profiles and their influences on the quality of life for individuals with schizophrenia, have been increasingly emphasized by the literature [7]. In terms of the conventional dopaminergic antipsychotics, SGA are generally preferred over FGA due to their overall more tolerable side effect profile, particularly decreasing extrapyramidal symptoms [7]. As LAI treatment options for antipsychotics have become available, they have become increasingly popular options due to their association with increased medication compliance as well as decreased hospitalizations and durations of hospitalizations for patients [6,7]. Adherence, side effects, and functional outcomes are important factors alongside symptom alleviation/reduction to take into greater consideration when augmenting aripiprazole LAI with additional antipsychotics. Although not recommended by major psychiatric organization guidelines, the use of dual LAI combinations of aripiprazole and an FGA has been documented to be effective in mitigating the symptoms of schizophrenia [14,15]. The FGAs fluphenazine, thiothixene (LAI no longer commercially available in the United States of America), and perphenazine all have LAI options that could potentially be used to augment aripiprazole LAI. However, due to the relatively similar K_i_ values of aripiprazole with these FGAs and its documented potential to decrease the effectiveness of other dopaminergic antipsychotics in treating symptoms of schizophrenia, targeting other mechanistic pathways of schizophrenia would likely be preferable for most patients [15]. Nevertheless, given the heterogeneity in study designs and outcome measures across the current literature, drawing definitive conclusions about the impact of LAI augmentation strategies on quality of life remains difficult. Further studies that integrate quality of life and functional outcomes are warranted to refine the current research.

Additionally, prior studies have indicated that approximately 47% of patients discontinue aripiprazole LAI within five years of initiating treatment, with 4.7% of patients citing poor tolerability and 14.1% citing ineffectiveness of the medication for discontinuing aripiprazole [6]. Patients continuing aripiprazole LAI after five years had an 88.5% reduction in the mean number of psychiatric hospitalizations (1.57 to 0.18, *p* < 0.001) and a 90% reduction in mean length of hospitalizations (103 to 10 days, *p* < 0.0001) compared to 5 years pre-aripiprazole LAI initiation [6]. In contrast, those that discontinued exhibited more negative outcomes and showed only a 29.9% reduction in admissions over 5 years (after switching from aripiprazole LAI to another LAI) [6]. Given the generally good tolerability of aripiprazole LAI, treatment augmentation has value as a method of improving medication effectiveness, which was the most commonly cited reason for medication discontinuation in the study [6]. Since oral clozapine and XT can be used to target different pathways than aripiprazole, these potential combinations of medications can theoretically be relatively safe, tolerable, and effective.

### 4.3. Future Directions

Further clinical trials are required before conclusive statements can be made about potential combinations for APP, in part due to the limited and conflicting existing literature, but also because there are many investigational compounds that have yet to be approved for the market [26]. Current pathways being explored include monoamine, glutamate, acetylcholine, cannabinoid receptors, enzyme inhibition, ion channel modulation, and mixed receptor modulation [26]. XT is the first approved drug that primarily targets the muscarinic pathway for the treatment of schizophrenia, with drugs targeting the same pathway like emraclidine, a selective M4 positive allosteric modulator, currently undergoing clinical trials [26,29]. With almost one in three patients having refractory/treatment-resistant schizophrenia, there is a rising need for the exploration of alternative pathways to target for the treatment of schizophrenia [21]. In the meantime, currently approved medications for the treatment of schizophrenia should undergo further clinical trials to study patient reactions to switching between antipsychotics of the same/different generation, differing drug classes, and for the efficacy and side effect profiles of antipsychotic polypharmacy. The current literature is highly contested, with many studies reporting conflicting findings for adverse event risks and clinical efficacy [14,15]. As in this narrative review, clinicians and researchers must speculate to some extent based on theoretical mechanisms and tailor their medication management for patients with schizophrenia based on their clinical judgement, potentially turning to the unorthodox option of APP as a last resort for treating refractory schizophrenia. Future clinical trials and studies must be rigorous and attempt to avoid bias as much as possible, as noted in some systematic reviews the limited sample sizes, difficulty screening out biased studies, and the relatively small number of references discussing LAI APP make it impossible to draw definitive conclusions about this subject [14].

## 5. Limitations

### 5.1. Methodological Limitations

As this is a narrative review, no formal analysis of the data from prior studies in the literature was conducted. This, in combination with the overall limited sample size of the literature on APP with aripiprazole LAI, prevents the conclusive determination of risks and side effect profiles of APP involving aripiprazole LAI. The quality and level of bias in the existing literature, which mainly consists of case reports or case series that lack control groups or standardized outcome measures, further hinder researchers’ ability to compare side effect profiles, safety, and efficacy of dual LAI APP [14]. For instance, differences in the duration of dual LAI use, diagnostic criteria, baseline patient characteristics, and concurrent medications may contribute to different outcomes, thereby reducing the validity of the current literature.

### 5.2. Literature-Based Limitations

Although the K_i_ database allows for the comparison of binding capacities of various antipsychotics for the DRD2 while controlling for factors such as the hot-ligand, species, and source that were used, its value is limited by a lack of peer-reviewed sources of binding capacities for many antipsychotics and by the lack of consistency in the hot ligands and species used in the studies within the database. Additionally, product vendor websites provide the binding capacities for aripiprazole metabolites without stating their methodology, binding ligands, species, or sources. With significant variations in K_i_ values noted between publications with the same authors, the authors elected to not include the reported K_i_ values of these product vendors as their lack of reporting on which hot-ligands, species, and sources used was deemed too significant of a confounding variable.

The literature is also not well-equipped to inform the long-term risk and benefits assessments of these treatments. Future studies will need to follow dual-LAI patients for longer periods of time as the existing literature does not contain long-term clinical trials or enough case reports/series that have followed the clinical courses of patients long enough to allow for a conclusive determination of the risks and benefits of long-term dual LAI treatment [13,14]. These concerns may include cumulative side effects, long-term metabolic risks, receptor desensitization, and treatment adherence. However, it may be worthwhile for researchers to conduct meta-analyses of existing data to examine the acute risks associated with dual LAI use, but this would require particular attention to the small sample sizes and significant variations in biases in the current literature [14].

Another limitation is the underrepresentation of non-Western populations in the current literature, which may affect the generalizability of findings to broader clinical settings. Additionally, there is a lack of neuroimaging and pharmacogenomic data correlating receptor occupancy and individual treatment response in the context of dual LAI regimens, limiting mechanistic understanding [3,4,5]. The absence of data on patient-reported outcomes, such as quality of life, functional recovery, and subjective side effect burden, also limits the ability to assess the full clinical impact of dual LAI use. The development and approval of alternative-mechanism antipsychotics like Xanomeline-trospium also give rise to the possibility of future adjunctive therapies with muscarinic agonists. However, this remains entirely speculative as the clinical efficacy of such treatment regimes is wholly unknown due to the absence of clinical trials.

With these limitations in mind, future reviews would benefit from being structured as systematic reviews or meta-analyses, especially when considering investigational drugs that are currently undergoing clinical trials which could address the existing limitations of the literature by providing more concrete data [26].

## 6. Conclusions

The existing literature on antipsychotic polypharmacy involving aripiprazole LAI is currently limited in both sample size and control for biases. This restricts the ability of researchers to draw firm conclusions on the efficacy, safety, and long-term consequences of antipsychotic polypharmacy involving aripiprazole LAI. However, there is a trend in newer publications towards a more favorable view of the use of APP and dual LAI treatment, particularly in patients with schizophrenia or schizoaffective disorder, than that present in older literature [15]. However, it should be noted that the prevalence of adverse side effects remains disputed in the literature, with multiple studies and references presenting conflicting information. Additional research exploring this topic is required to better understand the mechanisms behind schizophrenia and hopefully improve outcomes for patients experiencing treatment-resistant psychotic disorders or difficulty with adherence to antipsychotic medications.

This narrative review found that the most commonly used FGAs for dual FGA LAI were haloperidol decanoate, zuclopenthixol decanoate, and flupenthixoldecanoate while the most commonly used SGAs for dual SGA LAI were aripiprazole monohydrate and paliperidone palmitate [14]. The use of antipsychotic polypharmacy may provide better clinical effects than treatment augmentation with other drug classes, like benzodiazepines, which can carry additional risks, including dependency and cognitive blunting, though this advantage remains contested in the literature [15]. In cases where D2 antagonism or partial agonism fails to produce adequate symptom control, alternative targets may warrant exploration. In particular, recent developments like XT offer potential future directions for adjunctive or alternative treatments, although robust clinical data are not yet available.

Moreover, this narrative review underscores the critical need for the employment of a Precision Psychiatry approach when considering antipsychotic polypharmacy involving long-acting injectable aripiprazole. Given the complexity of schizophrenia and schizoaffective disorders, future research must focus on identifying patient-specific biomarkers and pharmacogenetic profiles that can allow for personalized treatment. As antipsychotic development begins to incorporate novel mechanisms such as muscarinic and glutamatergic modulation, it becomes increasingly essential to reassess the standard dopaminergic-centric treatment framework when managing medications for patients with schizophrenia. If targeting the dopaminergic pathway does not appear to be effective in a patient, the authors believe that clinicians should consider targeting alternative pathways for schizophrenia, such as the muscarinic pathway, though it is more indirect.

The risks of adverse effects following dual LAI use, and APP more broadly, in patients with psychosis are not fully understood, as multiple case reports indicated no increase in risk of adverse events, while some reviews found the opposite [13,14,15]. Until more high-quality clinical trials become available, clinicians must carefully consider the individual needs and circumstances of patients experiencing psychosis and calibrate treatments in coordination with patients to optimize clinical outcomes and patient quality of life.

## Figures and Tables

**Figure 1 diseases-13-00274-f001:**
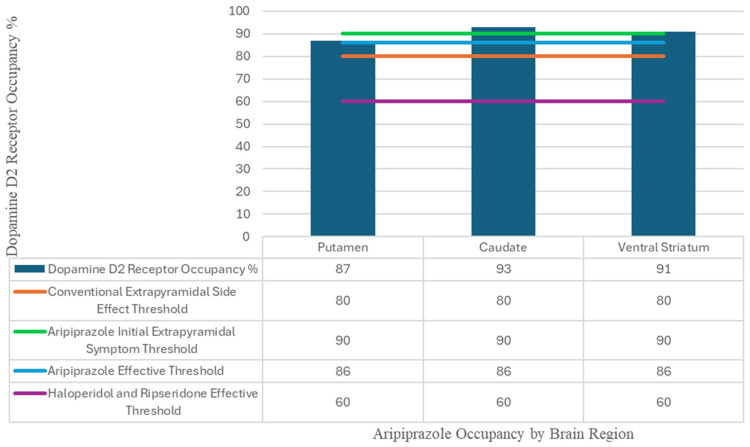
Visualization of the dopamine D2 occupancy percentages for aripiprazole in multiple regions of the brain [3]. Minimum dopamine D2 occupancy percentages for clinical effect were included for haloperidol and risperidone [3]. Percentage values where extrapyramidal symptoms began to appear in patients were also included for haloperidol, risperidone, and aripiprazole [3].

**Table 1 diseases-13-00274-t001:** Binding affinities and other considerations of various antipsychotics.

Antipsychotic	Average K_i_ Value (nM)	K_i_ Value Range	Metabolism	Notes/Considerations
Fluphenazine	0.54	0.540000 ^a,b^	CYP2D6	More susceptible to CYP2D6 inhibitors than other antipsychotics. ^d^
Pimozide	0.65	0.650000 ^b^	CYP2D6	Moderate sensitivity to CYP2D6. ^d^
Thiothixene	1.015	0.630000–1.400000 ^a,b^	-	-
Trifluoperazine	1.3	1.300000 ^b^	-	-
Perphenazine	1.4	1.400000 ^b^	-	Multiple pathways of clearance, limited impact of CYP induction/inhibition on serum levels. ^d^
Aripiprazole	2.455	0.660000–3.300000 ^a,b,c^	CYP2D6, CYP3A4	Manufacturer recommends twofold dose increase when administered with inducers like carbamazepine and a two-fold dose reduction with inhibitors such as fluoxetine, quinidine, or ketoconazole. ^d^
Chlorpromazine	3	2.000000–4.000000 ^a,b^	CYP1A2	Recommended to avoid smoking tobacco as it induces CYP1A2. ^d^
Haloperidol	3	2.000000–4.000000 ^a,b^	Primarily CYP2D6, CYP3A4, and CYP3A5	Has multiple pathways, limiting impact of CYP induction/inhibition on serum levels. ^d^
Risperidone	5.7	4.900000–6.500000 ^a,b^	CYP2D6	Serum levels may rise with CYP2D6 inhibitors like fluoxetine or paroxetine, will increase to a lesser degree with bupropion. ^d^
Ziprasidone	6.85	4.000000–9.700000 ^a,b^	Primarily through glutathione and aldehyde oxidase with minor contribution from CYP3A4	Inhibitors and inducers of CYP only cause modest effects in serum levels. ^d^
Sertindole	9.1	9.100000 ^b^	-	-
Thioridazine	10.5	10.000000–11.000000 ^a,b^	-	Multiple pathways of clearance, limited impact of CYP induction/inhibition on serum levels. ^d^
Loxapine	11	10.000000–12.000000 ^a,b^	-	Multiple pathways of clearance, limited impact of CYP induction/inhibition on serum levels. ^d^
Olanzapine	53	34.000000–72.000000 ^a,b^	CYP1A2	Avoid strong inhibitors or inducers of CYP1A2 as well as smoking tobacco. ^d^
Molindone	63	63.000000 ^b^	-	-
Clozapine	343.5	256.000000–431.000000 ^a,b^	Primarily CYP1A2 and CYP3A4, some CYP2D6	May require reduction of one-third dose with fluvoxamine or ciprofloxacin, smoking may require twofold increase in dose with 30–40% reduction of dose in settings where smoker is unable to smoke. ^d^
Quetiapine	410.5	254.000000–567.000000 ^a,b^	CYP3A4	Manufacturer recommends up to a fivefold increase in dose with CYP3A4 inducers like carbamazepine and a reduction to one-sixth with inhibitors like voriconazole or ritonavir. ^d^

All K_i_ values were sourced from the PDSP K_i_ database, https://pdsp.unc.edu/rothlab/publications.php, accessed on 14 May 2025. The following search terms were used in the database: receptor—dopamine D2; hot ligand—3H-NMSP; species—human; source—cloned. If multiple K_i_ values were found, their K_i_ values were provided as a minimum-maximum range and as an average. K_i_ values of <30 nM are high affinity, K_i_ values between 30–300 nM are moderate affinity, and K_i_ values >300 nM are low affinity. Abbreviations: K_i_ is inhibitory constant, nM is nanomolar, CYP is Cytochrome P450, CYP2D6 is Cytochrome P450 2D6, CYP3A4 is Cytochrome P450 3A4, CYP1A2 is Cytochrome P450 1A2. ^a^ [8]. ^b^ [11]. ^c^ [12]. ^d^ [9].

**Table 2 diseases-13-00274-t002:** Muscarinic Receptors.

Receptor	G Protein Subtype	Function	Regions of the Brain Mainly Found in	Dopamine Pathway(s) Affected
M1	G_q_	Excitatory	Cerebral Cortex, basal ganglia	Nigrostriatal pathway more than Mesolimbic pathway
M2	G_i/o_	Inhibition	Nucleus Basalis, hippocampus, basal ganglia	Mesolimbic pathway more than Nigrostriatal pathway
M3	G_q_	Excitatory	Cerebral cortex, basal ganglia	Mesocortical pathway
M4	G_i/o_	Inhibition	Basal ganglia and cortex	Nigrostriatal pathway more than Mesolimbic pathway
M5	G_q_	Excitatory	Hippocampus, substantia nigra, and VTA	Mesolimbic pathway more than Mesocortical pathway

M1, M2, M3, M4, and M5 are subtypes of muscarinic acetylcholine receptor. Gq and Gi/o are heterotrimeric G protein alpha subunits.

## Data Availability

No new data was created or analyzed in this study. Data sharing is not applicable to this article.

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
