# Peer review of "Considerations for Augmenting Aripiprazole Long-Acting Injectables with Other Antipsychotics: A Mini-Review"

_diseases, 2025, doi:10.3390/diseases13080274_

Round 1

Reviewer 1 Report

Comments and Suggestions for Authors

Dear Editor

Thanks for your kind invitation.

I was delighted to review the article for your prestigious journal.

It is a true honor that you allow me to make some suggestions.

I hope this will help the Authors achieving a better article:

TITLE

Dear Authors: congratulations. I work in a Treatment Resistant Schizophrenia setting and one of my favorite medication schemes, even popular among the patient and carer, is the combination of monthly aripiprazole LAI plus clozapine daily oral at night time.

AUTHORS

Please, avoid generic electronic addresses like @tmamd.com.

Provide institutional electronic addresses for all authors.

Explain the meaning of all acronyms, such as “M.D.” and “CA”.

Be consistent in the use of dots for acronyms. I the Authors use dots for “M.D.” they should also use dots for “CA.”

Delete the word “and” after “Tina Allee”.

ABSTRACT

Reading this section didn’t allow me to understand what kind of article is this.

Is this a perspective? A review? An original research work?

Besides, there is a huge pitfall: The Authors, in the Abstract only mentions a lot of FGA but only one SGA generation antipsychotic, which is clozapine. What about all the other SGA drug, like Olanzapine?

INTRODUCTION

Please, specify which “United States” are the Authors referring to? United States of Mexico? United States of America?

Please, explain all acronyms, such as “D2”, “5-HT2A”, “5HT1A”.

METHODS
I couldn’t find a Methods section. Please, provide one.

Instead of a Narrative Review or a Mini Review, we need a Systematic Review.

Please, provide us a Systematic Review of the literature, instead.

Please, follow PRISMA and PROSPERO guidelines.

https://www.prisma-statement.org/

https://www.crd.york.ac.uk/prospero/

RESULTS

I couldn’t find a RESULTS section. Please, provide one.

I admit all sections before DISCUSSION should be headed by a RESULTS heading.

Again, explain all acronyms, such as “D3”, “DRD2s”, “D2LR”, “emax”, “P2D6”, “CYP”, “CYP3A4”, “BMS-337041”, “IC50”, “Kd”, “e.g.”, “etc.”, “nM”, “FGA”, “H1”, “CI”, “RR”,

I would suggest some consistence while using abbreviations.

For example: “5-HT1A” has no lower characters for “1A”, but I found lower characters fo “Ki”.

Another example: “e.g.” and “etc.” is written with dots, while other acronyms are not.

Please, explain the meaning of “e.g.” and “etc.” before using the abbreviations.

Please, whenever using non-English words use italics. For example, when explain “exempli gratia (eg)” and “et cetera (etc)”, but also “id est (ie)”.

Please, avoid stigmatizing writing style. Do not ever “schizophrenic” but “patients with schizophrenia”.

Please, provide information regarding all combinations possible: Aripiprazole + FGA versus Aripiprazole + SGA, versus Aripiprazole + TGA. The last hypothesis is not mentioned at all.

Please, provide a new RESULTS section after using PRISMA and PROSPERO guidelines.

DISCUSSION

Please, explain all acronyms, such as “M1”, “NMDA”, “GABA”, “EMERGENT”, “ARISE”,

The Authors do not need to explain again the acronyms “SAE” that was already explained before.

Some Authors consider xanomeline - trospium chloride the first of fourth generation antipsychotic. That would be awkward because the acronym would be FGA, raising confusion with First generation anti-psychotics. As xanomeline – trospium as no Dopamine effects how would the Authors classify this new drug?

CONCLUSIONS
Please, avoid stigmatizing writing style. Do not ever “schizophrenic” but “patients with schizophrenia”.

I would rather use an “h” after “t” for both FGA “zuclopenthixol” and “flupenthixol”. Just because of coherence and elegance.

ABBREVIATIONS
The list very incomplete. I would delete it. In case the Journal demands it from the Authors there are many Abbreviations and Acronyms that need to be included in the section.

TABLES

Please, explain all acronyms used in the Tables, at the bottom of the Table. Especially in Table 2 that has no information at the bottom.

Please, the search methodology should not appear at the bottom of the Table 1, but instead in a proper METHODS section.

The Table is the first time that important SGA drugs are mentioned in the full article. That makes no sense. The Authors should have mentioned before SGA drugs in the ABSTRACT and in the INTRODUCTION

Where are the other TGA, such as brexpiprazole, cariprazine and lumateperone?

Before Table 2 I got the impression that there are already LAI versions for ziprasidone, lurasidone, and cariprazine. Is that true? Do not forget that LAI is the acronym for long acting injectable. I am not aware of LAI version for SGA besides Risperidone and Paliperidone. I dream about Olanzapine and Clozapine LAI versions, but the pharma industry seems to show little interest in these options.

Author Response

Manuscript ID: diseases-3795034 “Considerations for Augmenting Aripiprazole Long-Acting Injectables with Other Antipsychotics: A Narrative-Review”

The authors would like to thank all of the peer reviewers for providing their time and expertise. We have found that your suggestions and comments were very helpful in improving the quality and rigor of this review. Please see below our responses for each of your suggestions. We have copied your suggestions verbatim and responded to each suggestion in bold.

Peer Reviewer 1:

TITLE

Dear Authors: congratulations. I work in a Treatment Resistant Schizophrenia setting and one of my favorite medication schemes, even popular among the patient and carer, is the combination of monthly aripiprazole LAI plus clozapine daily oral at night time.

AUTHORS

Please, avoid generic electronic addresses like @tmamd.com.

Provide institutional electronic addresses for all authors.

We have updated the author emails to the following:

Jonathan Shaw 1*, Ethan Kim 2, MD, Emily Ton 1, Charles Lai 1, Peter Bota 1, Tina Allee, MD 3*

Affiliation 1: School of Medicine, California University of Science and Medicine, Colton CA

Affiliation 2: Loma Linda University Health, Loma Linda CA

Affiliation 3: Psychiatry, College Medical Center, Long Beach CA

*Correspondence: jonathan.shaw@md.cusm.edu; tallee@collegemedicalcenter.com

Note: Dr Allee’s email ends in .com but this is her institutional email to her faculty position at the College Medical Center Psychiatry residency program.

Explain the meaning of all acronyms, such as “M.D.” and “CA”. - MD stands for Doctor of Medicine, the equivalent of MBBS (Bachelor of Medicine, Bachelor of Surgery) elsewhere in the world. CA stands for California, a state in the United States of America. 

Be consistent in the use of dots for acronyms. I the Authors use dots for “M.D.” they should also use dots for “CA.” - We have removed the periods from the MD degrees in the title page

Delete the word “and” after “Tina Allee”. – We have made this change

ABSTRACT

Reading this section didn’t allow me to understand what kind of article is this.

- Clarified by indicating this is a narrative review, changed title to reflect this

Is this a perspective? A review? An original research work?

- Clarified by indicating this is a narrative review, changed title to reflect this

Besides, there is a huge pitfall: The Authors, in the Abstract only mentions a lot of FGA but only one SGA generation antipsychotic, which is clozapine. What about all the other SGA drug, like Olanzapine?

- According to the UNC Ki database, none of the SGA antipsychotics bind stronger than aripiprazole to the dopamine D2 receptor, therefore they were not listed. Olanzapine was found to have a Ki of 53, which indicates it does not have as strong as a binding capacity as aripiprazole which has a Ki of 2.455

INTRODUCTION

Please, specify which “United States” are the Authors referring to? United States of Mexico? United States of America?

- The United States Food and Drug Administration is the official name of the US FDA. We have indicated elsewhere that the United States is the United States of America.

Please, explain all acronyms, such as “D2”, “5-HT2A”, “5HT1A”. – We have added explanations for these acronyms

METHODS

I couldn’t find a Methods section. Please, provide one.

- Added methods section

Instead of a Narrative Review or a Mini Review, we need a Systematic Review.

- This is a narrative review, we cannot retroactively convert this into a systematic review

Please, provide us a Systematic Review of the literature, instead. - See above

Please, follow PRISMA and PROSPERO guidelines. - See above

https://www.prisma-statement.org/

https://www.crd.york.ac.uk/prospero/

RESULTS

I couldn’t find a RESULTS section. Please, provide one.

- Added a results section and collated all sections before the discussion to the results

I admit all sections before DISCUSSION should be headed by a RESULTS heading. - See above

Again, explain all acronyms, such as “D3”, “DRD2s”, “D2LR”, “emax”, “P2D6”, “CYP”, “CYP3A4”, “BMS-337041”, “IC50”, “Kd”, “e.g.”, “etc.”, “nM”, “FGA”, “H1”, “CI”, “RR”,

- The authors have added explanations for these acronyms/abbreviations, though “BMS-337041” is just the name of the metabolite and there is nothing to explain.

I would suggest some consistence while using abbreviations.

For example: “5-HT1A” has no lower characters for “1A”, but I found lower characters fo “Ki”. - Changed 5-HT1A to subscript for 1A to maintain consistency

Another example: “e.g.” and “etc.” is written with dots, while other acronyms are not.

- The authors believe that this is how these are written in standard English

Please, explain the meaning of “e.g.” and “etc.” before using the abbreviations.

- These are standard abbreviations of Latin phrases that are commonly used in scholarly discourse, the authors believe that these abbreviations are so universal that they do not require explanation in the manuscript for the sake of brevity.

Please, whenever using non-English words use italics. For example, when explain “exempli gratia (eg)” and “et cetera (etc)”, but also “id est (ie)”.

- https://www.mdpi.com/authors/layout#_bookmark15

Per the official MDPI Style Guide, under subsection “Italics”: Foreign words do not need to be highlighted or italicized, including Greek/Latin terms, such as i.e., e.g., etc., et al., vs., ca., cf., in vivo, ex vivo, in situ, ex situ, in vitro, in utero, ad hoc, in silico, ab initio, vice versa, and via. Authors may choose to use italics for purposes of emphasis or where a term is being defined.

Please, provide information regarding all combinations possible: Aripiprazole + FGA versus Aripiprazole + SGA, versus Aripiprazole + TGA. The last hypothesis is not mentioned at all.

- The authors have added aripiprazole plus TGA in the discussion section. It was not explored by the reference Cipolla et al., 2024 as it only examined dual LAI therapy and the authors of that systematic review did not indicate that they did a separate analysis for each individual combination of LAI. Aripiprazole is the only TGA with an LAI option currently approved for the market.

Please, provide a new RESULTS section after using PRISMA and PROSPERO guidelines.

- As this is narrative review and not a systematic review, we have not filled out a PRISMA checklist. However, we did collate multiple sections into a new results section

DISCUSSION

Please, explain all acronyms, such as “M1”, “NMDA”, “GABA”, “EMERGENT”, “ARISE”,

The Authors do not need to explain again the acronyms “SAE” that was already explained before.

- We have changed this

Some Authors consider xanomeline - trospium chloride the first of fourth generation antipsychotic. That would be awkward because the acronym would be FGA, raising confusion with First generation anti-psychotics. As xanomeline – trospium as no Dopamine effects how would the Authors classify this new drug?

- As Bristol-Myers Squibb refers to it as, “First-In-Class Muscarinic Agonist” we believe that if we were to abbreviate it, then it would be “First Generation Muscarinic Antipsychotic” or FGMA. Due to no other federally approved medications existing in the same class, we have not included this in the manuscript.

CONCLUSIONS

Please, avoid stigmatizing writing style. Do not ever “schizophrenic” but “patients with schizophrenia”.

- Thank you for bringing this to our attention, we have changed our writing style to avoid stigmatization of patients with schizophrenia.

I would rather use an “h” after “t” for both FGA “zuclopenthixol” and “flupenthixol”. Just because of coherence and elegance.

- We have changed the spelling of these FGA to match your suggestion

ABBREVIATIONS

The list very incomplete. I would delete it. In case the Journal demands it from the Authors there are many Abbreviations and Acronyms that need to be included in the section.

- We have reviewed the list of abbreviations and acronyms, updating them as needed. We believe that the journal would prefer for an acronym list to be there

TABLES

Please, explain all acronyms used in the Tables at the bottom of the Table. Especially in Table 2 that has no information at the bottom.

- We have added this to the Tables

Please, the search methodology should not appear at the bottom of the Table 1, but instead in a proper METHODS section.

- We have added this to the methods section

The Table is the first time that important SGA drugs are mentioned in the full article. That makes no sense. The Authors should have mentioned before SGA drugs in the ABSTRACT and in the INTRODUCTION

-  Due to the 250 word limit of the abstract, we were not able to elaborate on other SGA. We had also omitted them from the abstract because none of their Ki values were stronger than aripiprazole’s. We do mention paliperidone and ziprasidone briefly prior to the table, but the majority of the manuscript does focus on introducing the intricacies of aripiprazole because that is the antipsychotic this narrative review is focused on. We saved discussion about other antipsychotics, mainly those with stronger Ki values than aripiprazole, for the discussion section.

Where are the other TGA, such as brexpiprazole, cariprazine and lumateperone?

- Their Ki values were not available on the UNC Ki database, so we could not determine what ligand and sources were used in measuring their Ki values. We omitted them from the table as a result. We have also included a discussion of their viability as a replacement for aripiprazole LAI in the discussion section. We believe that, as aripiprazole is the only TGA with an LAI option currently approved, that if antipsychotic polypharmacy were to be attempted with a TGA then aripiprazole would be more convenient to use for patients, likely improving medication compliance.

Before Table 2 I got the impression that there are already LAI versions for ziprasidone, lurasidone, and cariprazine. Is that true? Do not forget that LAI is the acronym for long acting injectable. I am not aware of LAI version for SGA besides Risperidone and Paliperidone. I dream about Olanzapine and Clozapine LAI versions, but the pharma industry seems to show little interest in these options.

- Our apologies for the confusion, the authors had copied the list of antipsychotics medication participants had previously been on from the clinical trial website. The phrasing made it seem like there were LAI versions of ziprasidone, lurasidone, and cariprazine but in reality, there are no LAI versions of these antipsychotics. The LAI statement was meant for Risperidone/Paliperidone and Aripiprazole. The authors have removed the LAI statement for clarity.

Reviewer 2 Report

Comments and Suggestions for Authors

This narrative review addresses the clinically relevant issue of augmenting aripiprazole LAI with other antipsychotic agents in cases of insufficient response, particularly in acute settings. The manuscript provides a clear overview of aripiprazole’s pharmacodynamic and pharmacokinetic characteristics, including its unique receptor profile and partial agonism at D2 and 5-HT1A receptors. It also presents a detailed discussion on dopamine D2 receptor binding affinities (Ki values), and the theoretical rationale for combining aripiprazole LAI with other antipsychotics. The review includes an overview of the sparse literature on dual LAI regimens and explores the potential role of alternative mechanisms, such as muscarinic modulation. The discussion is appropriately cautious regarding the limitations of current evidence but remains largely theoretical.

The introduction provides an adequate rationale for the review, and the mechanistic discussion is well-developed.T the link between pharmacological mechanisms and real-world treatment outcomes is somewhat weak. There is a need to connect theoretical receptor data with meaningful clinical endpoints such as patient functioning and quality of life.

Sections on augmentation strategies and dual LAI
The review presents a structured summary of the limited and heterogeneous literature on antipsychotic polypharmacy, especially the combination of LAI formulations. The discussion includes both pharmacodynamic and safety considerations. Nonetheless, the review remains focused on receptor binding and pharmacokinetics, while paying only minimal attention to outcomes that matter to patients, such as subjective well-being, social functioning, or recovery trajectories. This is briefly acknowledged in the limitations section, but not sufficiently developed.

To strengthen this important point, I recommend citing the recent systematic review by Sampogna et al., which evaluated the impact of antipsychotic drugs on the quality of life of patients with schizophrenia. This paper is highly relevant because it documents how different antipsychotics affect subjective and functional outcomes beyond symptom reduction. Its inclusion would reinforce the argument that the current literature on LAI augmentation and antipsychotic polypharmacy is lacking not only in methodological robustness, but also in clinically meaningful outcome measures. This reference should be cited also to support the call for future studies that integrate quality of life, functioning, and recovery into the assessment of augmentation strategies.

Discussion and conclusion
The discussion of muscarinic-targeting agents adds conceptual richness to the review, although the speculative nature of these approaches should be more clearly distinguished from clinically available strategies. The final section summarizes the key points appropriately, but would benefit from a more explicit call for research focused on patient-reported and functional outcomes.

This is a well-written and useful narrative review. It would benefit from the inclusion of evidence linking pharmacological strategies to real-world outcomes. The inclusion of the following reference is therefore strongly encouraged:

Sampogna G, Di Vincenzo M, Giuliani L, Menculini G, Mancuso E, Arsenio E, Cipolla S, Della Rocca B, Martiadis V, Signorelli MS, Fiorillo A. A Systematic Review on the Effectiveness of Antipsychotic Drugs on the Quality of Life of Patients with Schizophrenia. Brain Sci. 2023;13(11):1577. doi:10.3390/brainsci13111577

This addition would support the authors’ statement on the need to go beyond receptor occupancy and symptom reduction, and help contextualize the importance of evaluating augmentation strategies in terms of patient-centered outcomes.

Author Response

Peer Reviewer 2:

The introduction provides an adequate rationale for the review, and the mechanistic discussion is well-developed.T the link between pharmacological mechanisms and real-world treatment outcomes is somewhat weak. There is a need to connect theoretical receptor data with meaningful clinical endpoints such as patient functioning and quality of life.

- We have expanded on this point in the discussion section, dedicating a subheader to quality of life

Sections on augmentation strategies and dual LAI

The review presents a structured summary of the limited and heterogeneous literature on antipsychotic polypharmacy, especially the combination of LAI formulations. The discussion includes both pharmacodynamic and safety considerations. Nonetheless, the review remains focused on receptor binding and pharmacokinetics, while paying only minimal attention to outcomes that matter to patients, such as subjective well-being, social functioning, or recovery trajectories. This is briefly acknowledged in the limitations section, but not sufficiently developed.

To strengthen this important point, I recommend citing the recent systematic review by Sampogna et al., which evaluated the impact of antipsychotic drugs on the quality of life of patients with schizophrenia. This paper is highly relevant because it documents how different antipsychotics affect subjective and functional outcomes beyond symptom reduction. Its inclusion would reinforce the argument that the current literature on LAI augmentation and antipsychotic polypharmacy is lacking not only in methodological robustness, but also in clinically meaningful outcome measures. This reference should be cited also to support the call for future studies that integrate quality of life, functioning, and recovery into the assessment of augmentation strategies.

- The authors have added a subheader for quality of life in patients in the discussion section, we have also included a separate future directions subheader too.

Discussion and conclusion

The discussion of muscarinic-targeting agents adds conceptual richness to the review, although the speculative nature of these approaches should be more clearly distinguished from clinically available strategies. The final section summarizes the key points appropriately, but would benefit from a more explicit call for research focused on patient-reported and functional outcomes.

- See above

This is a well-written and useful narrative review. It would benefit from the inclusion of evidence linking pharmacological strategies to real-world outcomes. The inclusion of the following reference is therefore strongly encouraged:

Sampogna G, Di Vincenzo M, Giuliani L, Menculini G, Mancuso E, Arsenio E, Cipolla S, Della Rocca B, Martiadis V, Signorelli MS, Fiorillo A. A Systematic Review on the Effectiveness of Antipsychotic Drugs on the Quality of Life of Patients with Schizophrenia. Brain Sci. 2023;13(11):1577. doi:10.3390/brainsci13111577

This addition would support the authors’ statement on the need to go beyond receptor occupancy and symptom reduction, and help contextualize the importance of evaluating augmentation strategies in terms of patient-centered outcomes. - See above

Reviewer 3 Report

Comments and Suggestions for Authors

This is a detailed narrative review entitled "Considerations for Augmenting Aripiprazole Long-Acting Injectables with Other Antipsychotics: A Mini-Review". The review is well written and merits publication. However several issues should be resolved before it is finally accepted:

  1. Title and Abstract: The title is precise and clearly communicates the topic of the paper. The abstract is informative, but the sentence “While, on average, fluphenazine, pimozide, thiothixene…” is too long and should be broken down for clarity. Please also summarize key findings more explicitly stating the knowledge gap this review aims to address.
  2. Introduction: The introduction provides a solid overview of aripiprazole’s pharmacological profile and clinical uses. However, the transition from the pharmacologic rationale to the research objective is somewhat disjointed. Please add a clearer statement of the objective near the end of the introduction, and highlight the clinical relevance of LAI augmentation more explicitly.
  3. Methodology: There is a strong and well-referenced summary of pharmacodynamics and pharmacokinetics of aripiprazole. However, heavy reliance on dense numerical data (Ki values, occupancy percentages) could overwhelm readers. Please, use instead a figure or graphical summary to visually display receptor affinity and occupancy comparisons across antipsychotics.
  4. Clinical Evidence and Polypharmacy Discussion: The review makes a meaningful effort to synthesize literature on dual LAI and antipsychotic polypharmacy (APP).

However, some statements (e.g., regarding clozapine, XT trials) are not adequately contextualized with statistical robustness or practical applicability. Please differentiate more clearly between theoretical receptor-based rationale and empirical clinical evidence and indicate study types (case reports vs. RCTs) when discussing findings.

  1. Discussion of Alternative Pathways (e.g., Muscarinic): The inclusion of muscarinic and glutamatergic pathways is a valuable expansion. However, some mechanistic pathways are overly speculative given the current state of clinical evidence. Please, clarify that discussions of NDMC and XT are exploratory and not standard clinical recommendations. Indicate explicitly which agents are FDA-approved vs. investigational.
  2. Limitations: The authors peovide a comprehensive and honest appraisal of limitations. Please consider separating methodological limitations (e.g., narrative design, small sample sizes) from literature-based limitations (e.g., lack of Ki data, sparse long-term trials) into subheadings for clarity.
  3. Conclusions: The authors summarize key points well and emphasizes the need for individualized treatment. However, they overstate the “support” for APP based on currently limited and conflicting evidence. Please, rephrase to emphasize cautious optimism and reinforce the need for more high-quality data before broad clinical recommendations can be made.
  4. References: Citations are relevant and from high-impact journals. Some references (e.g., PDSP database) are not clearly integrated or cited in the narrative. Please, ensure all database-derived values are clearly linked to a source or figure to maintain credibility and traceability.
  5. General comments: There are redundancies in expression (e.g., “psychotic patients with schizophrenia”) and typographic inconsistencies (e.g., spacing before parentheses, missing punctuation). A thorough line-editing pass would improve fluency and professionalism. Consider reworking some long paragraphs for better readability.
Comments on the Quality of English Language

Editing is needed by professional editors. 

Author Response

Peer Reviewer 3:

This is a detailed narrative review entitled "Considerations for Augmenting Aripiprazole Long-Acting Injectables with Other Antipsychotics: A Mini-Review". The review is well written and merits publication. However several issues should be resolved before it is finally accepted:

Title and Abstract: The title is precise and clearly communicates the topic of the paper. The abstract is informative, but the sentence “While, on average, fluphenazine, pimozide, thiothixene…” is too long and should be broken down for clarity. Please also summarize key findings more explicitly stating the knowledge gap this review aims to address.

- The authors have reworked the sentence for better clarity and explicitly stated what the knowledge gap is. However, the authors are limited by the 250 word limit for the abstract.

Introduction: The introduction provides a solid overview of aripiprazole’s pharmacological profile and clinical uses. However, the transition from the pharmacologic rationale to the research objective is somewhat disjointed. Please add a clearer statement of the objective near the end of the introduction, and highlight the clinical relevance of LAI augmentation more explicitly.

- We have highlighted the clinical relevance of LAI augmentation (in terms of potentially increased medication compliance and improved quality of life for patients) and have rewritten our objectives for clarity.

Methodology: There is a strong and well-referenced summary of pharmacodynamics and pharmacokinetics of aripiprazole. However, heavy reliance on dense numerical data (Ki values, occupancy percentages) could overwhelm readers. Please, use instead a figure or graphical summary to visually display receptor affinity and occupancy comparisons across antipsychotics. - The authors have added a figure to help visualize these values

Clinical Evidence and Polypharmacy Discussion: The review makes a meaningful effort to synthesize literature on dual LAI and antipsychotic polypharmacy (APP).

However, some statements (e.g., regarding clozapine, XT trials) are not adequately contextualized with statistical robustness or practical applicability. Please differentiate more clearly between theoretical receptor-based rationale and empirical clinical evidence and indicate study types (case reports vs. RCTs) when discussing findings.

- Clarified that the mechanism NDMC is based on theoretical receptor-based rationale, but its effects on cognition are backed up with results from Phase II clinical trials. Also added that the XT trials are RCTs.

Discussion of Alternative Pathways (e.g., Muscarinic): The inclusion of muscarinic and glutamatergic pathways is a valuable expansion. However, some mechanistic pathways are overly speculative given the current state of clinical evidence. Please, clarify that discussions of NDMC and XT are exploratory and not standard clinical recommendations. Indicate explicitly which agents are FDA-approved vs. investigational.

- Added short statement at the end of the mechanistic considerations subheader of the discussion section explicitly stating that the discussion of the mechanism of how NDMC and XT could augment aripiprazole LAI is theoretical and these agents are not FDA approved for augmenting existing dopaminergic antipsychotic therapy

Limitations: The authors peovide a comprehensive and honest appraisal of limitations. Please consider separating methodological limitations (e.g., narrative design, small sample sizes) from literature-based limitations (e.g., lack of Ki data, sparse long-term trials) into subheadings for clarity.

- Separated these into individual subheadings, added statement about how future reviews on this subject would benefit from being meta-analyses/systematic reviews given the fact that there are many investigational drugs currently undergoing clinical trials

Conclusions: The authors summarize key points well and emphasizes the need for individualized treatment. However, they overstate the “support” for APP based on currently limited and conflicting evidence. Please, rephrase to emphasize cautious optimism and reinforce the need for more high-quality data before broad clinical recommendations can be made.

- The authors have cited the reference which states that there is a trend of growing support in some researchers for APP, but also agree that this should be interpreted very cautiously due to the very limited and contradictory data currently available in the literature. We have rephrased the statement to “view in a more favorable light” and rephrased the following sentence to emphasize the limitations of the existing data.

References: Citations are relevant and from high-impact journals. Some references (e.g., PDSP database) are not clearly integrated or cited in the narrative. Please, ensure all database-derived values are clearly linked to a source or figure to maintain credibility and traceability.

- The authors contacted Dr. Bryan Roth and Estela Lopez of the UNC Ki database directly through email and were informed that they would like to be referenced with the URL to the Roth lab (which is the PDSP database), we have clarified that in the table and updated the reference list to reflect this.

General comments: There are redundancies in expression (e.g., “psychotic patients with schizophrenia”) and typographic inconsistencies (e.g., spacing before parentheses, missing punctuation). A thorough line-editing pass would improve fluency and professionalism. Consider reworking some long paragraphs for better readability.

- The authors have made edits to the manuscript to improve readability and the professionalism of the review. We hope that the quality of the language used in this review is suitable. Hopefully, the free English editing services that the journal provides for minor corrections will be suffice, but we would be open to more rigorous paid English editing services if need be.

Round 2

Reviewer 2 Report

Comments and Suggestions for Authors

Thank you very much for your valuable work

Reviewer 3 Report

Comments and Suggestions for Authors

All my comments have been succesfully addressed.